# Urban Sprawl and Health Outcome Associations in Sicily

**DOI:** 10.3390/ijerph16081350

**Published:** 2019-04-15

**Authors:** Vincenzo Restivo, Achille Cernigliaro, Alessandra Casuccio

**Affiliations:** 1Department of Health Promotion Sciences, Maternal and Infant Care, Internal Medicine and Medical Specialties (PROMISE), University of Palermo, Via del Vespro 131, 90127 Palermo, Italy; alessandra.casuccio@unipa.it; 2Department of Health Services and Epidemiological Observatory, Regional Health Authority, Sicilian Region, Via Mario Vaccaro 5, 90145 Palermo, Italy; achille.cernigliaro@regione.sicilia.it

**Keywords:** urban sprawl, sprawl index, working activity, urban health, cardiovascular, transportation

## Abstract

Urban sprawl has several negative impacts on the environment, the economy, and human health. The main objective of this work was to formulate and validate a sprawl/compactness index for Sicilian municipalities and evaluate its association with health outcomes. An ecological study was conducted with 110 municipalities in Sicily, Italy. Principal component analysis was adopted to create the sprawl/compactness Sicilian index, and linear regression analysis was used to evaluate the association between the sprawl index and health outcomes. More variance of the new sprawl index was explained by the working factor, followed by density, surface extension, and land use mix. When validating the index, we found that public transportation had an inverse relation with sprawl increase (*p* < 0.001), and private transportation was directly related to the increase in sprawl (*p* < 0.001). After controlling for the Sicilian socio-economic deprivation index and overall mortality, cardiovascular mortality was the only outcome directly associated with the increase in the sprawl index (odds ratio = 0.0068, *p* < 0.001). Urban sprawl has to be monitored in Sicily over time to understand the evolution of the urbanization phenomenon and its relationship with health outcomes such as cardiovascular mortality. The use of the sprawl index should help policymakers define the necessary strategic aspects and actions to improve human health and quality of life in cities through a multi-sectorial approach.

## 1. Introduction

Urban sprawl is a phenomenon that can be visually perceived in the landscape and is characterized by low population density, automobile dependence, and single-use land zoning. The opposite of sprawl, has been described as a more efficient way of using the land, including urban development that is compact (spatial arrangement) and dense (low land uptake per person) [1]. Several definition of urban sprawl can be found in the literature, but a systematic evaluation showed that most of these definitions have three dimensions in common: expansion of urban areas equal to the strong growth of urban settlements; the scattering of settlement areas, which involves evaluating how widely dispersed the buildings are within the landscape; and low-density development, assessed as high land uptake per person [1]. According to these characteristics, the more an area is built in a given landscape (amount of built-up area), the more dispersed this built-up area is in the landscape (spatial configuration), and the higher the uptake of the built-up area per inhabitant or employed (lower use intensity in built-up areas), the higher the degree of urban sprawl [2]. Urban sprawl has several negative impacts on the environment (water, energy, air, climate, land, and biodiversity), the economy, and human health [3]. Increased air pollution [4,5] is the most extensively investigated environmental impact of urban sprawl. A study conducted in 2017 analyzed satellite-based measurements of nitrogen dioxide (NO_2_) and a global data set of 1,274 urban areas to explore the relationship between urban form and air quality in large and small cities (100,000−500,000 people). Results showed that increased urban contiguity, circularity, and vegetation metrics are significantly associated with lower urban NO_2_ concentrations (*p* = 0.008). The impervious surfaces metric has an insignificant effect (*p* = 0.19) on urban NO_2_ concentration [6].

A further consequence of urban sprawl is the growing consumption of energy related to transport. Evidence from cities around the world showed that higher energy consumption rates are associated with lower urban densities, characteristic of sprawling environments [7]. Compact urban developments with higher urban densities are more energy efficient. Cities with high transport energy use per capita were located in the USA followed by Canada, Australia, and Western Europe [7].

Several studies indicated that urban sprawl has a negative impact on human health. For example, increased automobile use has negative effects on health, resulting in higher levels of inactivity and a higher frequency of obesity, in addition to the higher number of pedestrian injuries [8,9,10]. Although prior studies showed that environmental factors can influence health, the studies only examined single dimensions of location. The sprawl construct incorporates more complex components of an area’s overall accessibility as opposed to primarily relying on single factors such as proximity or density [11]. Sprawl measures should include proximity, density, walkability, street network connectivity and safety, presence or lack of community centers, the neighborhood mix of jobs, homes, and service types. These factors, when considered together, may explain why location strongly influences risk for health.

In the United States, where urban sprawl was first noted in the 1970s, principal component analysis (PCA) and 2010 cross sectional data for large urbanized areas were used to operationalize compactness/sprawl in each of four dimensions: development density, land use mix, activity centering, and street accessibility. Higher values of the index represent greater compactness and lower values of sprawl [12]. Generalizing across overall large urbanized areas in the United States, urban sprawl was shown to increase between 2000 and 2010, but only slightly. Atlanta was the most sprawling city and San Francisco was the most compact large urbanized area [12]. Several indexes have been validated and tested to measure urban sprawl. In China a study evaluated urban land expansion using high-resolution Landsat Thematic Mapper and Enhanced Thematic Mapper data. In this study China’s urban land increased by 80.8% during 1990–1995 and 19.2% during 1995–2000. Case studies of the 13 mega cities showed that urban expansion had been largely driven by demographic change, economic growth, and changes in land use policies and regulations [13]. Another technique used in Europe to measure the degree of urban sprawl was the weighted urban proliferation (WUP) method. The overall value of WUP in Europe (32 countries combined) increased from 1.56 urban permeation units (UPU)/m^2^ in 2006 to 1.64 UPU/m^2^ in 2009 (1.7% per year). Urban sprawl increased in most European countries by more than 1% per year, and by even more than 2% per year in many countries between 2006 and 2009. A wide range of urban sprawl values were found: low values were obtained for large parts of Scandinavia (<1 UPU/m^2^) and high (>4 UPU/m^2^) to very high (>6 UPU/m^2^) values for Western and Central Europe [1]. With a greater level of detail, urban sprawl was found to be most pronounced in wide rings around city centers, along large transport corridors, and along many coastlines, particularly in Mediterranean countries [1]. Despite the awareness that the continuous increase in building sector land use leads to huge costs for the community and has a strong impact on the quality of the environment, built up areas have rapidly grown in Italy. In 2011, the built up area in Italy reached 6.7% of the total area, which is an increase of 9% since 2001 [14]. In detail, both compact and sprawled built up area represented 26% of the total, whereas the sprawled built up area alone accounted for 14%. This built-up area included more large residential areas and coastal sites, and the remaining was attributed to small-scale settlements predominantly for commercial use [14].

Sicily is a larger developed administrative region in Italy; the spread of built-up area has never been evaluated with an urban sprawl metric, nor has its relationship with health consequences. The main objective of this work was to produce and to validate a sprawl/compactness index for Sicilian municipalities and evaluate its association with health outcomes.

## 2. Material and Methods

### 2.1. Study Design and Sample

An ecological study was conducted on Sicilian municipalities. Sicily is one of 20 Italian administrative regions. It is the fourth most populous Italian region, with 5,026,989 residents in 2018. Sicily has an economy mainly based on services for tourists and agriculture. It also has a low number of industries on the national level [14]. This study was conducted among 110 municipalities: those with more than 10,000 residents in 2011. The rationale for limiting the sample was to produce a homogeneous sprawl/compactness score for large urban areas. Small urban Sicilian areas are different in terms of built environmental characteristics than large urban areas. In small areas, neighboring land uses are similar, guaranteeing more accessibility. According to the Italian Institute of Statistics (Istat), congestion levels are low [14]. Small urban areas have lower densities than large urban areas because they have lower rent curves and lower land prices.

The sprawl/compactness Sicilian index for each municipality incorporates four dimensions of sprawl: work, density, surface extension, and land use mix.

Economic and demographic factors account for more variance of cost and accessibility in public transportation. Among several factors, sprawl indexes are associated with active commuting, but its association can vary in size and direction for different transportation types and for higher- and lower-income areas [15]. The working activity factor incorporates the following variables: percentage of working active people, percentage of working people in commercial sector, and percentage of working people in tourism sector. Istat to categorize working activity used the Italian classification of work (CP2011), derived from updating the CP2001 version and adapted to the innovations introduced by the International Standard Classification of Occupations (Isco08) [16,17].

In Italy, urbanization land use is closely linked to urban building and its extension, so a density factor was developed [14]. The density factor is evaluated using building area per km^2^, population density, municipal area covered by continuous residential buildings, and municipal area covered by sparse residential buildings. The building area was evaluated by satellite images as covered buildings, areas isolated from streets or empty spaces or other constructions, that had one or more accesses, and one or more autonomous stairs [18].

The surface extension factor includes variables related to the municipal surface extension of urban area and the proportion of municipal urban area in a province. The total extent of the Italian territory (the sum of the surface area of the 8,092 Italian municipalities as of the 2011 Italian census) was 302,070.8 km^2^. This value was determined using the cartographic archives available to Istat, updated as of the 2011 general census, and was assessed using geographic information system (GIS) tools [19]. Sicily, with an area of 25,832 km^2^ (8.6% of the national area), was the largest Italian administrative region, subdivided into nine administrative provinces. Among the administrative provinces, Palermo was the largest at 5009 km^2^, followed by Catania at 3574 km^2^, and Messina at 3266 km^2^ [19].

Land zone uses are also included in most lists of factors affecting sprawl development patterns. Mixed and integrated land uses represent pedestrian-friendly, transit-oriented, and smart growth patterns [15]. The land use mix was used as a variable, including municipal altitude and municipal area covered by broad-leaved woods. Istat, within the 2011 Italian census, assessed the altitudes of the Italian municipalities using GIS methodology and adopting the Zonal statistics as table algorithm of ARCGIS 10.1 (Esri Italia, Rome, Italy) [18].

To evaluate health outcomes in this study, we used Sicilian mortality rate. The death data was sourced from the Sicilian Register of Causes of Death (ReNCaM) [20]. The Sicilian ReNCaM estimates mortality indicators considering: sex, province and municipality of residence, year of death, province and municipality of death, year of birth, age at the time of death, and the principal cause of death. The causes of death were codified using the ninth revision of the International Classification of Diseases, Traumas and Causes of Death (ICD-9CM). In this analysis, mortality rate was grouped by anatomical apparatus using the following ICD9-CM codes: cardiovascular (390–459), respiratory (460–519), neoplastic (140–209 and 230–231), and overall mortality rate (001–009).

Another variable we used to control for health outcome was the already published Sicilian socioeconomic deprivation index [21,22]. The index was built for each municipality according to specific inequality dimensions: education level, family composition, unemployment, and house crowding. This index was divided by population quintiles: very high deprivation level, high deprivation level, medium deprivation level, low deprivation level, and very low deprivation level.

Our study was conducted using anonymous and unidentifiable data according to Italian privacy law.

### 2.2. Statistical Methods

PCA was used to derive the sprawl index of Sicilian municipalities. The principal component selected to represent one dimension was the component that captured the largest share of common variance among the variables. Principal factor extraction was used followed by orthogonal rotation, which allows for determination of the correlation among the factors. A four-factor model was fitted, examining factor loadings for each model.

The sprawl phenomenon had a consistently recognized association with automobile dependence [2]. A correlation analysis of the sprawl index and transportation outcomes was conducted to validate the index using Pearson correlation. After controlling for other relevant influences, it is expected that compact urbanized areas have relatively high transit, walkability, and short drive times to work.

To evaluate the association between the sprawl index and health outcomes, we used linear regression analysis. The dependent variable was logged to be normally distributed and hence properly modelled with regression analysis. All analysis was performed using Stata v14.2 software (StataCorp LLC, College Station, TX, USA) and for all analyses, a *p*-value < 0.05 was considered to indicate significance (two-tailed).

## 3. Results

Table 1 shows the results of the factorial analysis identifying four factors. The working factor, with an eigenvalue of 4.57, indicates that one factor accounts for more than one-third of the total variance in the dataset. All component variables load positively on the working factor. The eigenvalue for the density factor was 3.77. The density factor accounts for more than one-quarter of the total variance in the dataset. All component variables load positively on the density factor. The surface extension factor had an eigenvalue of 1.78, which is an explained variance of over 1/10. Both component variables load positively on the surface extension factor. The eigenvalue of mix land using factor was 1.31, which is an explained variance of less than 1/10.

Although the working factor was the most explaining dimension of sprawl, there is no rationale for assigning different weights to the four factors. Each affects the accessibility or inaccessibility of development patterns no more than others. Depending on the values, each factor can move along the continuum from sprawl to compact development. Therefore, the sprawl index was created containing predictions for factors scored by the Bartlett method [23].

Table 2 presents overall compactness scores and individual component scores for the 10 most sprawling and the 10 most compact large urbanized areas. Using these metrics, Palermo was the most sprawling city and Riposto was the most compact.

The relationships between sprawl and travel outcomes were used to validate the urbanized area sprawl measures. Table 3 shows the correlation analysis between the sprawl/compactness index and the public or private transportation rate for Sicilian municipalities, extracted from the 2011 Italian census, used to validate the index. Public transportation had an inverse relationship with sprawl increase (β coefficient = –0.185, *p* < 0.001); private transportation had a direct relationship with the increase in sprawl (β coefficient = 0.187, *p* < 0.001).

Table 4 show results of univariate and multivariate linear regression analysis of urban sprawl with health-related outcomes. After controlling for the Sicilian socio-economic deprivation index, mortality for neoplasia, respiratory disease, and the overall mortality, only cardiovascular mortality had a direct association with the increase in the sprawl index, with odds ratio (OR) = 0.0068, *p* < 0.001.

## 4. Discussion

This study developed an overall measure of compactness/sprawl for Sicilian municipalities to allow for future longitudinal comparisons. The factor explaining more of the variance in the sprawl index variables was the working factor. This highlights the impact of active commuting on urban sprawl, which can vary in size and direction for different transportation types and for higher-income and lower-income areas [2]. Urban sprawl may reduce community-level social capital by increasing commute times and creating geographic barriers between areas where people work, live, and shop [24]. According to this evaluation, the orographic characteristics of the Sicilian landscape, socioeconomic conditions, and working modalities could have considerably influenced urbanization policies. The consequences of sprawl are not restricted to suburban neighborhoods, but may accrue within an entire economic region, including areas in the region that are not themselves sprawling [25]. Sprawl may increase traffic across a central city and its surrounding suburbs as people commute to and from work.

The sprawl index developed in Sicily showed a higher level of consistency with the Italian report that compared urbanization density and dimension of municipalities in 2011 [14]. According to these figures, Catania was the Sicilian city with the largest urbanized area, followed by Palermo. Messina was the city with the highest urbanization density, followed by Palermo. The Sicilian sprawl index considered urbanization density in its factorial analysis, and Palermo was the Sicilian city with the highest density factor value, followed by Catania and Messina. Larger municipalities showed higher sprawl indexes than smaller cities in Sicily. This could be related to low-urbanization policies developed around city centers in larger cities and the lack of in-filling policies, using the gaps within existing built-up areas such as unused sites or brownfields.

In Sicily, urban sprawl is associated with cardiovascular mortality rate after controlling for other mortality rates and age. Another study assessed the relationship between urban sprawl and coronary artery disease endpoints, such as myocardial infarction and coronary artery disease death in a female population in the United States. In this study, women living in areas with a lower degree of compactness (or higher urban sprawl) were at significantly higher risk of coronary artery disease events or death than those living in more compact areas, after adjusting for obesity and weekly calorie expenditure [26]. Fewer opportunities to walk in more compact areas, perhaps due to fewer sidewalks and nearby walking destinations, would explain the association between mortality rate and urban sprawl. The association of urban sprawl with cardiovascular disease and sedentary lifestyle showed a greater risk for non-communicable diseases linked with lifestyle. Some studies showed a statistically significant link between the elements of the built environment and the risk of obesity, suggesting that some built environments may be more “obesogenic” than others [8,9]. In the United States, data from the Behavioral Risk Factor Surveillance System were collected from 350,000 adults to investigate the relationship between health risk behaviors and chronic disease [10]. The results showed, after controlling for individual and country level variables, that both original and new compactness indexes were negatively related to body mass index, obesity, heart disease, high blood pressure, and diabetes. Even if the causal mechanism was not assessed, low levels of active travel may be partially responsible for this result. Urban sprawl could have a negative impact on other health outcomes, like mental health. An Italian study assessed this relationship by looking at variations in antidepressant prescriptions depending on specific dimensions of the built environment (urban density, land use mix, green areas, public services, and accessibility through public transport) [27]. The effect of both variables was reported in men and women over 50 years old.

Another study showed that urban sprawl impacts life expectancy. This study was conducted in the United States, where the ranking for life expectancy decreased from 11th place in 1980 to 21st in 2006, to explain the connections between urban sprawl and life expectancy for metropolitan counties in the United States in 2010 [28]. After controlling for socio-demographic characteristics, the authors found that life expectancy was significantly higher in compact counties than in sprawling counties. A life expectancy increase of 3% was found with a doubling of the compactness index (increase by 100%). This corresponded to a 2.5-year increase in life expectancy for Americans who had a current expectancy of 78 years.

The main limitation of our study was the small sample size of the study. This bias can be balanced considering that, in Italy, there have been no other studies analyzing urban sprawl indexes. Another limitation is that mortality rate can be affected by existing illness unassociated with sprawl design. Even if we corrected for other mortality rates, the covariates reduced the risk attributed to the cross-sectional design.

## 5. Conclusions

This study created an urban sprawl metric for Sicily that should to be monitored over time to understand the evolution of the urbanization phenomenon and to help plan transport and settlement development, as population density is related to accessibility and the cost effectiveness of public transport. The use of the sprawl index should help designers, local governments, public bodies, policy makers, and all professionals working at local health agencies to define the strategic aspects and actions to improve human health and quality of life in cities through a multi-sectorial approach to public health promotion policies.

## Figures and Tables

**Table 1 ijerph-16-01350-t001:** Variables loading of sprawl/compactness index for Sicilian municipalities.

Urban Sprawl Factors	Factor Loading
**Working Factor**	
Percentage of working people by province	0.946
Percentage of working people in commercial sector by province	0.934
Percentage of working people in touristic sector	0.955
Eigenvalue	4.57
Explained variance	30.8%
**Density Factor**	
Municipal population density	0.851
Municipal density of building surface	0.917
Municipal area covered by continuous residential building	0.759
Municipal area covered by sparse residential building	0.882
Eigenvalue	3.77
Explained variance	25.4%
**Surface Extension Factor**	
Municipal surface	0.816
Proportion of provincial surface covered by municipal area	0.800
Eigenvalue	1.78
Explained variance	12.0%
**Mixed Land Use Factor**	
Municipal altitude on sea level	0.604
Mean of municipal broad-leaved woods	0.818
Eigenvalue	1.31
Explained variance	8.8%

**Table 2 ijerph-16-01350-t002:** Sprawl/compactness score for most 10 sprawling and 10 most compact Sicilian cities.

City	Sprawl/Compact Index	Working Factor	Density Factor	Surface Extension Factor	Mixed Land Use Factor
**10 Most Sprawling Cities**					
Palermo	12.4	9.1	3.1	0.9	–0.6
Ragusa	10.5	5.0	−0.2	6.0	–0.3
Caltanissetta	9.1	4.6	–0.1	4.6	–0.1
Messina	8.7	6.7	0.5	1.8	–0.4
Siracusa	8.2	6.3	0.5	2.2	–0.8
Modica	7.4	4.0	0.1	3.7	–0.3
Gela	6.6	4.2	–0.1	3.1	–0.8
Noto	6.4	0.6	–0.2	6.4	–0.3
Enna	6.3	3.0	–0.2	3.5	0.1
Catania	4.7	3.8	0.4	1.2	–0.7
**10 Most Compact Cities**					
Ravanusa	–0.9	0.1	–0.3	–0.2	–0.6
Scordia	–1.1	–0.1	0.3	–0.5	–0.7
Casteldaccia	–1.1	–0.2	–0.1	–0.4	–0.4
Grammichele	–1.1	–0.2	–0.1	–0.4	–0.3
Belmonte Mezzagno	–1.2	–0.3	–0.3	–0.5	–0.1
Santa Teresa di Riva	–1.3	–0.1	0.1	–0.6	–0.6
Santa Flavia	–1.4	–0.3	0.3	–0.6	–0.8
Palagonia	–1.4	–0.2	–0.2	–0.1	–0.7
Motta Sant’Anastasia	–1.5	–0.1	–0.4	–0.4	–0.6
Riposto	–1.5	–0.1	–0.1	–0.6	–0.8

**Table 3 ijerph-16-01350-t003:** Correlation between urban sprawl and transportation type.

Transportation Type	Sprawl Index	*p*-Value
Public	−0.185	<0.001
Private	0.187	<0.001

**Table 4 ijerph-16-01350-t004:** Univariate and multivariate regression analysis of sprawl compactness score and health outcome.

Health Outcome	Crude Odds Ratio (OR)	*p*-Value	Adjusted OR	*p*-Value
Overall mortality rate	0.0004	<0.001	–0.0019	0.428
Neoplasia mortality rate	0.0013	<0.001	–0.0004	0.941
Cardiovascular mortality rate	0.0011	<0.001	0.0068	0.008
Respiratory mortality rate	0.0054	<0.001	0.0016	0.729
Very high socio economic deprivation	3.8087	0.004	–0.3825	0.788

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
