# Peer review of "Urban Sprawl and Health Outcome Associations in Sicily"

_ijerph, 2019, doi:10.3390/ijerph16081350_

Round 1

Reviewer 1 Report

This manuscript formulated and validated a sprawl/compactness 13 index for Sicilian municipalities evaluating its association with health outcome. The impact of urban sprawl on health is an important and traditional topic, which is worthy exploring. However, there are some key concerns which the author/s still need to address.

1. The conceptual framework needs to be strengthened by engaging with existing studies and theories, why the four dimensions (work, density, surface extension and land use mix) can be formed as index for evaluating urban sprawl? What is the main contribution of this work?

2. A related point is how you selected the indicators/factors. For example, why the factors—percentage of working people in commercial sector, percentage of working people in touristic sector can reflect the working factor?? I suggest the author(s) to elaborate more on the factor selection whether they’re appropriate.

3. Given that the international Journal Environmental Research and Public Health caters to an international readership, this can also help readers who not familiar with Italy to understand Sicily’s situation by explaining why Sicily More importantly what implications does Sicily have for the other European cities? Perhaps it would be more helpful if the author/s can be briefer about the description of Sicily.

4. The format of citations and references should be further checked and unified.

Author Response

Reviewer 1

This manuscript formulated and validated a sprawl/compactness 13 index for Sicilian municipalities evaluating its association with health outcome. The impact of urban sprawl on health is an important and traditional topic, which is worthy exploring. However, there are some key concerns which the author/s still need to address.

1. The conceptual framework needs to be strengthened by engaging with existing studies and theories, why the four dimensions (work, density, surface extension and land use mix) can be formed as index for evaluating urban sprawl? What is the main contribution of this work?

Answer: We thank the Reviewer to improve quality of manuscript. Considering his observations was added the following paragraph in Introduction section “Several studies indicated that urban sprawl has a negative impact on human health. For example, increased automobile use has negative effects on health, resulting in higher levels of inactivity and a higher frequency of obesity in addition to the higher number of pedestrian injuries [8–10]. Although prior studies showed that environmental factors can influence health, the studies often only examined single dimensions of location. The sprawl construct incorporates more complex components of an area’s overall accessibility as opposed to primarily relying on single factors such as proximity or density [11]. Sprawl measures should include proximity, density, walkability, street network connectivity and safety, presence or lack of community centers, and the neighborhood mix of jobs, homes, and service types. These factors, when considered together, may explain why location strongly influences risk for health.”

2. A related point is how you selected the indicators/factors. For example, why the factors—percentage of working people in commercial sector, percentage of working people in touristic sector can reflect the working factor? I suggest the author(s) to elaborate more on the factor selection whether they’re appropriate.

Answer: The selection of factors was conducted according to type of activities conducted more frequently in Sicily. One of the most productive activity was related to touristic services in Sicily. To make more clear the selection process the author included a description of Sicily in the method section.

3. Given that the international Journal Environmental Research and Public Health caters to an international readership, this can also help readers who not familiar with Italy to understand Sicily’s situation by explaining why Sicily More importantly what implications does Sicily have for the other European cities? Perhaps it would be more helpful if the author/s can be briefer about the description of Sicily.

Answer: We thank the Reviewer for make more clear to read the manuscript. Consequently, a brief description of Sicily was added to Methods section. “An ecological study was conducted on Sicilian municipalities. Sicily is one of the 20 Italian administrative regions. It is the fourth most populous Italian, with 5,026,989 residents in 2018. Sicily has an economy mainly based on services for tourists and agriculture. It also has a low number of industries on the national level [13].”

4. The format of citations and references should be further checked and unified.

Answer: The format of citation and reference was checked and unified.

Reviewer 2 Report

The use of English in the paper is far below the academic standard. There are many typos and grammatical errors.  For example, it is unclear about the term "the larger developed administrative region" in line 79.  Should it be "the largest"?  The authors should have the paper proofread by a professional English writer before submission.

The topic has been extensively studied in the literature.  However, the authors have not conducted a comprehensive review on the subject issues.  For example, there are many different measures or indicators of urban sprawl but the authors have ignored most of them.  I would suggest the authors to study more literature, particularly those focusing on East Asia.

The research methodology is problematic.  What's the meaning of a "restrospective study" (line 85)? Besides, how were the respondents in the survey sampled?  How was the survey conducted, face-to-face or postal?

A description of the respondent profile is needed.

Mortality rate represents just one of the many sides of health outcome.  It is not well justified to use a single dimension of health outcome for a high-quality scientific research.

What are the policy or practical implications of the research?

Author Response

Reviewer 2

Reviewer: The use of English in the paper is far below the academic standard. There are many typos and grammatical errors. For example, it is unclear about the term "the larger developed administrative region" in line 79. Should it be "the largest"? The authors should have the paper proofread by a professional English writer before submission.

Answer: The paper had a high level editing by the journal editing service. .

Reviewer: The topic has been extensively studied in the literature.  However, the authors have not conducted a comprehensive review on the subject issues.  For example, there are many different measures or indicators of urban sprawl but the authors have ignored most of them.  I would suggest the authors to study more literature, particularly those focusing on East Asia.

Answer: Even if there are several urban sprawl indicators only a little of them were monitored during time to evaluate change in urban sprawl. The author selected for their work only urban sprawl metrics that were evaluated during time. We did’nt find any assessment during time of urban sprawl in Asian Countries even if they use several metrics for different relationship.

Reviewer: The research methodology is problematic.  What's the meaning of a "restrospective study" (line 85)? Besides, how were the respondents in the survey sampled?  How was the survey conducted, face-to-face or postal? A description of the respondent profile is needed.

Answer: This study was an ecological study because analyzed data from indicators belonging to Sicilian cities. So, no survey  was conducted, but we used available data of indicators of Sicilian cities.

Reviewer: Mortality rate represents just one of the many sides of health outcome.  It is not well justified to use a single dimension of health outcome for a high-quality scientific research.

Answer: We thank the Reviewer to give us the possibility to clarify some aspects. First of all mortality rate are the more robust health outcomes when a new topic was treated for the first time like in Sicily. Furthermore, in this study several mortality specific rates were used to explore the role of urban sprawl.

Reviewer: What are the policy or practical implications of the research?

The authors explained policy and pratical implication of the study in the Conclusion section where it was stated that: “This study created an urban sprawl metric for Sicily that should to be monitored over time to understand the evolution of the urbanization phenomenon and to help plan transport and settlement development, as the population density is related to accessibility and the cost effectiveness of public transport. The use of the sprawl index should help designers, local governments, public bodies, policy makers, and all professionals working at local health agencies to define the strategic aspects and actions to improve human health and quality of life in cities through a multi-sectorial approach to public health promotion policies.”

Reviewer 3 Report

dear authors,

thanks for your interesting paper. 

Here some comments:

Table 1, correct the formatting of the first line.

I would have made more explicit the subdata values for each individual indicators.

There is a relationship between your indicators and the city size (in terms of population) that require more attention.

Interesting the idea to investigate the link between sprawl and some health issues, but I would have added more control variables before conclude such a causal link, especially socio-economical of the individual/family, as well as local pollution and presence of industrial activities nearby. 

Author Response

Reviewer 3

Dear authors,

thanks for your interesting paper.

Here some comments:

Reviewer: Table 1, correct the formatting of the first line.

Answer: The authors changed the italics format of the first line of table 1.

Reviewer: I would have made more explicit the subdata values for each individual indicators.

There is a relationship between your indicators and the city size (in terms of population) that require more attention.

Answer: The authors clarified the role of city size with the following sentence on Discussion section: “Larger municipalities showed higher sprawl indexes than smaller cities in Sicily. This could be related to low-urbanization policies developed around city centers in larger cities and the lack of in-filling policies, using the gaps within existing built-up areas, such as unused sites or brownfields.”

Reviewer: Interesting the idea to investigate the link between sprawl and some health issues, but I would have added more control variables before conclude such a causal link, especially socio-economical of the individual/family, as well as local pollution and presence of industrial activities nearby.

Answer: We thank  the Reviewer for their stimulating comments but with the exception of the socio/economic level of population evaluated inside “Socio-economic deprivation index”, other control variables were not available by the current data with a communal level of detail. Therefore, other study should be conducted to evaluate “environmental pollution” and “presence of industrial activities nearby” were not available using current data for Sicily.

Round 2

Reviewer 1 Report

The authors have addressed all my concerns.

Author Response

The authors have no comment.

Reviewer 2 Report

Although the authors attempted to address my conments, I don’t think the authors treated my comments seriously.  In particular, there is a large body of longitudinal studies euthanasia the use of on urban sprawl measures or indicators but these previous studies have been ignored by this paper.  Examples of these previous works include

https://doi.org/10.1016/j.landusepol.2008.02.005

https://doi.org/10.1016/j.landurbplan.2006.03.008

https://doi.org/10.1579/0044-7447-34.6.450

Regarding the research design, the authors clarified that statistical analyses were conducted based on secondary data. However, where did the data come from? Were the sources reliable? Were the collected data validated?

Author Response

Reviewer 2

Although the authors attempted to address my conments, I don’t think the authors treated my comments seriously.  In particular, there is a large body of longitudinal studies euthanasia the use of on urban sprawl measures or indicators but these previous studies have been ignored by this paper.  Examples of these previous works include

https://doi.org/10.1016/j.landusepol.2008.02.005

https://doi.org/10.1016/j.landurbplan.2006.03.008

https://doi.org/10.1579/0044-7447-34.6.450

Answer: The authors evaluated all suggestions received and included the study that developed an index for urban sprawl. This study was conducted in China evaluating driving force of urban expansion as specified in the manuscript introduction “In China a study evaluated urban land expansion using high-resolution Landsat Thematic Mapper and Enhanced Thematic Mapper data. In this study China’s urban land increased by 80.8% during 1990–1995 and 19.2% during 1995–2000. Case studies of the 13 mega cities showed that urban expansion had been largely driven by demographic change, economic growth, and changes in land use policies and regulations [12].”

Regarding the research design, the authors clarified that statistical analyses were conducted based on secondary data. However, where did the data come from? Were the sources reliable? Were the collected data validated?

Answer: The authors already showed source of data and data analysis in material and methods section. In particular, the main sources of data were Italian Institute of Statistics and Register of Causes of Death.

Reviewer 3 Report

Thanks for now.

Author Response

The authors have no comment.